# Mathematical Estimation of the Energy, Nutritional and Health-Promoting Values of Multi-Layer Freeze-Dried Vegetable Snacks

Monika Janowicz, Agnieszka Ciurzyńska , Magdalena Karwacka , Jolanta Kowalska and Sabina Galus *

Department of Food Engineering and Process Management, Institute of Food Sciences,
Warsaw University of Life Sciences—SGGW, 02-787 Warsaw, Poland; monika_janowicz@sggw.edu.pl (M.J.);
agnieszka_ciurzynska@sggw.edu.pl (A.C.); magdalena_karwacka@sggw.edu.pl (M.K.);
jolanta_kowalska@sggw.edu.pl (J.K.)
* Correspondence: sabina_galus@sggw.edu.pl

**Abstract:** Nowadays, the popularity of snack foods is increasing due to the fast-paced lifestyle of society. Thanks to the prevailing trends related to a healthy lifestyle and organic food, the need to create new products is increasing, but also more and more attention is being paid to high nutritional value. The aim of the study has been to evaluate the energy, nutritional, and health-promoting value of freeze-dried vegetable-based products with hydrocolloids as structure forming additives. The research included mathematical estimation of the energy and nutrients content, as well as selected health-promoting components, such as vitamins and micro- and macro-nutrients. The calculation was based on tabular data of the nutritional values each components of the products. In addition, the quality of the bars has been assessed by means of the daily requirement and the nutritional quality index. The bars have proven to be characterized by high energy and nutritional and health-promoting value. The Index Nutritional Quality (*INQ*) indicator has shown that the tested products are incorrectly adjusted in terms of the content of nutrients in relation to the energy supplied. The broccoli bar has turned out to be the best option because it has the highest content of protein, fat, and all the relevant vitamins and minerals. Obtained results verified that tested snacks were not enough to cover daily intake of specific nutrients, but introducing such products to balanced diet may have beneficial influence on human health and well-being.

**Keywords:** freeze-dried vegetable snacks; energy value; nutritional value; health-promoting value

## 1. Introduction

The fast lifestyle that is developing nowadays and the lack of time for daily preparation of meals increases the popularity of snack consumption. In addition, the prevailing trends related to a healthy and ecological lifestyle make consumers' awareness about a balanced, pro-health diet rich in ingredients. For this reason, more and more attention is paid to the nutritional value of products, which includes their energy value and the content of nutrients and health-promoting ingredients. Unhealthy, high-calorie, traditional snacks, such as crisps, cookies, or chocolate bars are being replaced with low-energy products, but rich in bioactive ingredients. Vegetables, as a rich source of vitamins, minerals, and antioxidants, are undoubtedly critical for such products. Unfortunately, fresh vegetables are available seasonally and are susceptible to mechanical damage and environmental conditions. For this reason, various drying techniques are often used for the purpose of reducing the water content in the raw vegetables, which allows to obtain a product with a higher pro-health value [1,2].

Freeze-drying is the method that best maintains the nutrients in vegetables. Additionally, this technique allows the original structure and colour to be retained, making the product more attractive to the consumer. In order to create snacks, it is possible to combine

the freeze-drying method with the use of hydrocolloids, i.e., texturing substances that allow to create the desired texture of the product. Moulding, that is used on an industrial scale, for the production of chips, among others, ensures the right shape of snacks. As a result of the combination of those methods, freeze-dried bars based on vegetables may be produced. However, this technique is not widespread, so it is important to assess the energy, nutritional, and health-promoting value of those products [3–5].

*1.1. Characteristics and Methods of Producing Snacks Based on Fruits and Vegetables*

Snacks are most often associated with unhealthy and high-energy products. However, it is difficult to determine the exact impact of snacks on the body because they are different from ordinary meals [1]. There is no single, clearly established definition of a snack. This term can generally be described as eating or drinking food or drinks between main meals. In the related literature, most definitions are based on the time of a day, amount, location of food consumption, its type, and a combination of those factors [6].

Fruit and vegetables are most often used for producing a healthy snack due to the high content of nutrients. Additionally, such raw fruit and vegetables are susceptible to mechanical damage, environmental conditions, and microorganisms, which makes them difficult to keep fresh in this form. In addition, they are available seasonally, therefore all kinds of processing methods are used for producing various forms of snacks with high quality and attractive sensory value [7–9].

Dried fruit and vegetables in the form of chips, such as apples, strawberries, carrots, tomatoes, and beets account for undoubtedly the simplest form of pro-health snacks. They are not only fat-free products but also low in calories ones. Drying the raw fruit and vegetables to reach low water content results in the achievement of the desired texture characterised by high brittleness and crunchiness, which, for this type of product, is a quality indicator [4,10]. In addition, reducing the water content concentrates nutrients and allows to obtain a product proving the pro-health value that is higher than fresh fruit or vegetables have [11–13]. The most common method of producing fruit or vegetable chips is convective drying. Konopacka and Płocharski [14] assessed the quality and sensory attractiveness of crisps made from fruit and vegetables for direct consumption. The analysis covered snacks made of banana, pear, celery, and red beet, obtained by impregnating them in sugar solutions (some variants with the addition of spices) and subsequent convective. The sensory analysis has shown that consumers see high potential in such snacks. For all the tested samples, their appearance, taste, and crispness are highly esteemed. It also concluded that celery crisps are the most attractive product according to consumers. Kozak [15], on the other hand, assessed the texture of fruit and vegetable snacks available in the market. Apple, strawberry, tomato, pepper, carrot, and beetroot crisps were tested. It was concluded that the tomato crisps had the highest hardness, which could result in the lack of acceptance of the product by consumers. Carrot chips, on the other hand, had the best texture as they had the greatest crispness, which was an important quality indicator for snack products [4,16–18].

Due to the course of the freeze-drying process and its main feature, which is the removal of water by passing the liquid phase of the solvent, this process has many advantages, which makes it widely applicable. The main advantage of freeze-drying is that this process allows to remove almost 100% of the water from the dried product or leaves only a small amount of it. Such low water content in products facilitates long-term food preservation [17]. Examining the effect of convective drying and freeze-drying on the colour of fruit has shown that the freeze-drying process made it possible to maintain the colour similar to the original raw product. In the conducted consumer study, the taste, colour, and general quality of the lyophilisates were assessed better than the convection-dried product [19]. Analysing the influence of drying methods on the properties of dried Japanese quince chips Kowalska et al. [12] showed that freeze-drying was the best drying method as compared to the convective drying. The product was characterised by the lowest water activity and content, as well as the lowest colour changes as compared to product. Guine and Barroca [20],

while examining the effect of convective and freeze-drying conditions of pumpkin and green pepper, concluded that the tested parameters of quality, such as colour and texture, indicated a much better quality of freeze-dried samples than that of convection-dried one. The freeze-drying process causes smaller colour changes as an effect of a slight and much smaller decomposition of chlorophyll and other pigments, and non-enzymatic reactions during convective drying. Another advantage of freeze-drying is that freeze-dried products are easy to reconstitute by rehydration, which is very important. Jokić et al. [21], examining the effect of the drying method on wild asparagus rehydration, showed that rehydration characteristics of the freeze-dried samples were significantly better than in the case of convective drying and the lower was freezing temperature the higher was rehydration ratio of freeze-dried material. Assessing the content of bioactive ingredients and antioxidant properties of freeze-dried powders delivered from plants, the freeze-drying method allows for obtaining products rich in bioactive compounds: vitamin C, polyphenols, including anthocyanins and carotenoids, characterised by high antioxidant activity. Due to the above advantages of the freeze-drying process, it may be used for producing food for special nutritional purposes, convenience food and dietary supplements. In addition, it is a suitable method for processing vegetables and fruit [5,22–26].

Another group of snacks, quite popular and often consumed, are extruded products, such as corn crisps or puffs. Their main component is starch, which is the most important element in the extrusion method, as it is responsible for the formation of porous structure and brittleness typical of those products [27–29]. Due to the fact that corn crisps are quite poor in nutrients, they are more and more often enriched with other additives. The extrusion method allows for effective combination of raw cereal with aroma and oil fruit and vegetables, obtaining a product of good quality and durability [30]. Pęksa et al. [28] examined the quality of corn crisps enriched with amaranth flour, Jerusalem artichoke and pumpkin pulp. They recommended the use of amaranth seeds flour, which increased quality indicators. The effects of Jerusalem artichoke and pumpkin on the structure of the corn snacks were adverse and the industrial application was excluded. Gondek et al. [31] conducted the analysis of the physical and functional properties of high-fibre cereal and vegetable snacks. Cereal–zucchini–broccoli and cereal–pumpkin crisps produced by the extrusion and expansion method were tested. The values of water content and activity were obtained ensuring microbiological stability. Moreover, based on the density and porosity of the products, it was found that they were characterised by a delicate, brittle texture. It was also concluded that those products had adequate digestibility.

Producing fruit or vegetable bars is an innovative but not very popular method of obtaining a healthy snack. They may be single-layer or multi-layer ones. Hydrocolloids also have the ability to form gels, which maintain their shape and structure after water removal due to freeze-drying, and that allows to manufacture attractive for consumer products. Moreover, their ability to water binding and swelling capacity make them food components responsible for keeping the feeling of satiety for a long time. As was mentioned before, hydrocolloids are high molecular weight biopolymers that act in human's digestive system as dietary fibre. Thanks to referred properties of hydrocolloids, they are established as additives that can be used to create functional food [4,32,33]. Ahmad et al. [34] investigated the quality attributes of bars made from papaya and tomatoes with the addition of hydrocolloids, such as pectin or starch. In the sensory analysis, the taste, colour, texture, and aroma achieved a satisfactory level on a nine-point scale, the average value of the rating is 7. Moreover, it was found that the bars remained stable throughout the four-month storage. Ciurzyńska et al. [4,10] have developed multi-layer freeze-dried bars based on frozen vegetables. For the production of snacks, vegetables, such as potatoes, broccoli, peppers, corn, carrots, cauliflower, and green beans, were used. The bar variants also differed in the type of hydrocolloids used. In one case, it was sodium alginate and calcium lactate, and in the other, locust bean gum and xanthan gum. The obtained bars were characterised by a low water activity, which made them microbiologically stable. Additionally,

the organoleptic evaluation for all snack variants was positive, which confirmed the great interest in freeze-dried products of this type.

*1.2. Waste from the Food Industry in Terms of Applicability for the Production of Innovative, Nutritious Snacks*

Food waste and overproduction in rich countries, as well as malnutrition and hunger among the population of undeveloped regions, are among the current trends that characterize the modern food industry. The largest proportion of food waste is fresh produce, including fruit and vegetables, which that are susceptible to spoilage and, therefore, their shelf-life is short-lived. Actions undertaken by scientists as part of the fight against the problem are aimed at searching for the possibility of using available raw materials to produce easy-to-distribute food with extended shelf life and developing innovative production techniques, the implementation of which will result in a real improvement of the environment in terms of compliance with the principles of sustainable development.

It is estimated that one-third of food is wasted and lost as a result of unintentional or accidental activities in the world, i.e., about 1.3 billion tons of food produced both in the form of raw materials, semi-finished products, and ready-made products, more than half of which is thrown away by potential consumers after purchasing them [35]. In Europe, around 100 million tons of food is wasted annually, which is around 180 kg per person, most of which are plant-based products, while two-thirds would still be edible. The research shows that most food is wasted by households (around 42%), slightly less by producers (39%), around 14% is lost in the transport chain, while sales losses account for only 5% of the total amount of food wasted. In turn, data from the World Food Organisation (FAO) shows that the production of food waste in the countries of Europe and North America is 95–115 kg per person [36]. This is a dramatically larger amount if you compare it with the values for the countries of Africa or South East Asia, i.e., 6–11 kg per person. The poorer the country, the less is wasted. The consequences effect the whole environment. One ton of food stored after disposal represents 4.5 tons of harmful gases delivered to the atmosphere. Methane from rotting food is up to 20 times more harmful than carbon dioxide. Therefore, already in 2012, the European Parliament adopted a resolution calling for concrete measures to reduce food waste by half by 2025. One of the proposals is to release products in packages of various sizes to facilitate the purchase of the right amount, or to use full-value waste from the food industry to produce innovative, nutritious vegetable snacks.

The development of sustainable development indicators based on comparative analyses of real data regarding products, enterprises, and investments and their consistent application is necessary to reduce the degrading impact of human activity on the environment [37,38]. Food and its production constitute a source of many threats related to the concept of sustainable development, especially in terms of consumer health, limiting environmental degradation or highlighting social and economic inequalities. The type of production, processing, distribution, and consumption of food is also influenced by global and local trends, such as urbanisation, industrialisation, cultural and demographic changes, and climate change. Therefore, the cooperation of all links in the food chain is the key to improve the operation of the entire system and effective implementation of the sustainable development policy [9,35,37,39]. All those factors and principles of sustainable development should determine each stage of the design of food products and technological processes used for their production on an industrial scale. Optimalisation of this type should consist, inter alia, in the use of materials with a low environmental impact, the use of the so-called "clean" technologies in production and packaging processes and enabling reuse, recycling or ecological utilisation of waste [38,40].

The development of the agri-food industry in accordance with the principles of sustainable development is associated with the implementation of technologies that optimise the operation of processes at the social, economic, and environmental level. To this end, the aim is to reduce the harmful impact on the environment by increasing the efficiency of the use of raw products, energy, and other resources. Optimising the production processes

and composition of products brings about a change in their contribution to the quality of life of people in the field of health, education, and culture, at the employee, consumer, and community level. In economic terms, the implementation of the concept of sustainable development should result in increased productivity, the creation of cheap, high-quality products, and the creation of enterprises, and, hence, the creation of new jobs [9].

Taking into account the discussed issues, an attempt has been made to assess the energy, nutritional and health-promoting value of freeze-dried vegetable-based products with hydrocolloids. Presented investigations arise from the continuation of the research results published previously by Ciurzyńska et al. [4]. The scope of the work has included designing a methodology for estimating the energy, nutritional and health-promoting value of the prepared snacks based on hydrocolloid and vegetables in the form of bars, calculation of their energy value and the content of nutrients and health-promoting components. In addition, the quality of the bars has been assessed by covering the daily requirement and the nutritional quality index.

## 2. Materials and Methods

### 2.1. Research Material

The research input derived from formulations of freeze-dried products formed on the basis of vegetables obtained in laboratory conditions at the Department of Food Engineering and Process Management at the Warsaw University of Life Sciences, which are presented in Table 1. Technological methods used for the purpose of the research input and physical characteristics are described in details by Ciurzyńska et al. [4].

**Table 1.** Formulations of three-layer freeze-dried vegetable bars [4].

| Material | Layer | Composition (%) | | Material | Layer | Composition (%) | |
|---|---|---|---|---|---|---|---|
| 1—CPB$_A$ | I | Water<br>Carrot<br>Sodium alginate<br>Salt<br>Calcium lactate | 58.4<br>39.6<br>1.5<br>0.4<br>0.1 | 2—CPB$_G$ | I | Water<br>Carrot<br>Carob gum<br>Xanthan gum<br>Salt | 58.4<br>39.6<br>1.0<br>1.0<br>0.4 |
| | II | Water<br>Potato<br>Cauliflower<br>Sodium alginate<br>Salt<br>Calcium lactate | 58.4<br>20.0<br>19.8<br>1.5<br>0.2<br>0.1 | | II | Water<br>Potato<br>Cauliflower<br>Carob gum<br>Xanthan gum<br>Salt | 58.4<br>20.0<br>19.8<br>1.0<br>1.0<br>0.2 |
| | III | Water<br>Broccoli<br>Green bell pepper<br>Green string bean<br>Chive<br>Sodium alginate<br>Salt<br>Calcium lactate | 58.4<br>18.0<br>15.0<br>4.6<br>2.0<br>1.5<br>0.4<br>0.1 | | III | Water<br>Broccoli<br>Green bell pepper<br>Green string bean<br>Chive<br>Carob gum<br>Xanthan gum<br>Salt | 58.4<br>18.0<br>15.0<br>4.6<br>2.0<br>1.0<br>1.0<br>0.4 |
| 3—CPCB$_A$ | I | Water<br>Corn<br>Potato<br>Sodium alginate<br>Salt<br>Calcium lactate | 58.4<br>29.6<br>10.0<br>1.5<br>0.4<br>0.1 | 4—CPCB$_G$ | I | Water<br>Corn<br>Potato<br>Carob gum<br>Xanthan gum<br>Salt | 58.4<br>29.6<br>10.0<br>1.0<br>1.0<br>0.4 |
| | II | Water<br>Potato<br>Cauliflower<br>Leaf dill<br>Sodium alginate<br>Salt<br>Calcium lactate | 58.4<br>20.0<br>18.8<br>1.5<br>1.0<br>0.2<br>0.1 | | II | Water<br>Potato<br>Cauliflower<br>Leaf dill<br>Carob gum<br>Xanthan gum<br>Salt | 58.4<br>20.0<br>18.8<br>1.5<br>1.0<br>1.0<br>0.2 |
| | III | Water<br>Broccoli<br>Green bell pepper<br>Basil leaves<br>Sodium alginate<br>Salt<br>Calcium lactate | 58.4<br>25.0<br>13.6<br>1.5<br>1.0<br>0.4<br>0.1 | | III | Water<br>Broccoli<br>Green bell pepper<br>Basil leaves<br>Carob gum<br>Xanthan gum<br>Salt | 58.4<br>25.0<br>13.6<br>1.5<br>1.0<br>1.0<br>0.4 |

**Table 1.** *Cont.*

| Material | Layer | Composition (%) | | Material | Layer | Composition (%) | |
|---|---|---|---|---|---|---|---|
| 5—CYGP$_A$ | I | Water<br>Corn<br>Yellow string bean<br>Sodium alginate<br>Salt<br>Calcium lactate | 58.4<br>27.6<br>12.0<br>1.5<br>0.4<br>0.1 | 6—CYGP$_G$ | I | Water<br>Corn<br>Yellow string bean<br>Carob gum<br>Xanthan gum<br>Salt | 58.4<br>27.6<br>12.0<br>1.0<br>1.0<br>0.4 |
| | II | Water<br>Green bell pepper<br>Chive<br>Leaf dill<br>Sodium alginate<br>Salt<br>Calcium lactate | 58.4<br>20.0<br>19.6<br>8.0<br>1.5<br>0.4<br>0.1 | | II | Water<br>Green bell pepper<br>Chive<br>Leaf dill<br>Carob gum<br>Xanthan gum<br>Salt | 58.4<br>20.0<br>19.6<br>8.0<br>1.0<br>1.0<br>0.4 |
| | III | Water<br>Potato<br>Cauliflower<br>Sodium alginate<br>Salt<br>Calcium lactate | 58.4<br>20.0<br>19.6<br>1.5<br>0.4<br>0.1 | | III | Water<br>Potato<br>Cauliflower<br>Carob gum<br>Xanthan gum<br>Salt | 58.4<br>20.0<br>19.6<br>1.0<br>1.0<br>0.4 |

The freeze-dried snacks were developed using post-calibration frozen vegetable out-grades supplied by one of the participants of the project, local manufacturer Unifreeze sp. z o.o. (Miesiączkowo, Poland). Carrot, bell pepper, and potato were in the form of cubes (1 × 1 × 1 cm), beans were cut into 2 cm strings, and broccoli and cauliflower were fragmented to smaller pieces. Before processing, all vegetables and herbs used in the formulations were stored at the temperature of −18 °C. Two variants of hydrocolloids were used: sodium alginate with the addition of calcium lactate as a gelling initiator, as well as a mixture of xanthan gum and locust bean gum (carob gum). The first step of the snacks producing was to defrost the vegetables to make them soft enough to use a hand blender MSM87180 (Bosch, Gerlingen-Schillerhöhe, Germany). Carrot, broccoli, bell pepper, and cauliflower were thawed by convection for 30 min in water at a temperature of approximately 40 °C. Other vegetables, such as corn, potatoes, and green beans were first boiled in boiling water for about 20 min. Then, the ingredients were blended to a smooth mass, according to the formulations presented in Table 1. In each variant of snacks, vegetables and spices constituted 40% of all ingredients, the rest was water and hydrocolloids.

The amount of hydrocolloids added was determined experimentally having of the synergistic phenomenon. The proportion of sodium alginate was 1.5%, and calcium lactate 0.1%. The locust bean gum and xanthan gum were added in an amount of 2% in total, 1% each. The water temperature required to obtain a gel for the addition of sodium alginate and calcium lactate was about 40 °C, with the calcium lactate being dissolved in a small part of the water before adding it to the mixture. The water temperature for the locust bean gum mixture and xanthan gum was about 75 °C.

Such prepared batches were placed in layers in silicone moulds, freeze at −40 °C and freeze-dried at 30 °C for 48 h utilizing Alpha 1–4 LD plus freeze-drier (Martin Christ Gefriertrocknungsanlagen, Osterode am Harz, Germany).

### 2.2. Computational Methods

Estimation of the energy, nutritional, and health-promoting value of the freeze-dried multi-layer vegetable snacks was determined on the base of tabular data about nutritional value of components contained in the formulation of the products [41–43]. For the purpose of the study, it had been established that one serving of each snack was one freeze-dried bar (10 g) and each layer shared one-third of the total product weight. All calculations were made in duplicate using Excel 2019 (Microsoft, Redmond, Washington, DC, USA).

### 2.2.1. Calculating the Energy Value

The classical method, in which the net Atwater energy equivalents were used, was obtained by multiplying the gross Atwater equivalents by the degree of digestibility of individual nutrients: for 1 g of protein 4 kcal for 1 g of fat 9 kcal for 1 g of carbohydrates 4 kcal [39–41].

On the basis of the product composition and the data contained in the table of composition and nutritional value, the content of proteins, carbohydrates, and fats was determined for individual freeze-dried products based on vegetables. Then, the content of individual food ingredients was multiplied by the corresponding net Atwater energy equivalents according to the formula:

$$E = F \times 9 + P \times 4 + C \times 4 \tag{1}$$

where:

$E$—energy value (kcal/(100 g of product));
$F$—fat content (g/(100 g of product));
$P$—protein content (g/(100 g of product));
$C$—carbohydrate content (g/(100 g of product)).

### 2.2.2. Calculating the Nutritional Value

The method is based on tabular data of the nutritional values of individual vegetables from which the moulded products were prepared.

The weight of nutrients (proteins, fats, carbohydrates) of the individual raw materials products used to produce for producing the final product was calculated from the formula: The content of nutrients (proteins, fats, carbohydrates) of the individual raw materials used to produce the final product was calculated from the formula:

$$M_{SO} = \frac{m_p \times m_{SO}}{100}, \tag{2}$$

where:

$M_{SO}$—mass of a given nutrient (proteins, fats, carbohydrates) contained in the raw product used for production (g);
$m_p$—mass of raw product used for production (g);
$m_{SO}$—mass of a given nutrient in 100 g of raw product (proteins, fats, carbohydrates)—tabular value (g).

### 2.2.3. Calculating Pro-Health Value

The method is based on tabular data on the nutritional value of vegetables and on the basis of general knowledge about the pro-health value of vegetables. The calculations were made in duplicate.

It is believed that the health-promoting value of vegetables is determined in particular by the content of many vitamins (vitamin C, provitamin and in the form of carotenes—beta-carotene, niacin, vitamin E, folates). Vegetables are also a rich source of dietary fibre. In the human diet, 30–40% of the fibre comes from vegetables. They are also characterized characterised by a high content of mineral salts (iron, calcium, potassium, magnesium) and antioxidants (phenolic compounds, carotenoids) [39–41].

The content of ingredients determining pro-health value of individual raw materials used to produce the final product was estimated according to the formula:

$$M_{SO} = \frac{m_p \times m_{SO}}{100}, \tag{3}$$

where:

$M_{SO}$—mass of a given pro-health component contained in the raw product used for production (mg);
$m_p$—mass of raw product used for production (g);

$m_{SO}$—weight of a given pro-health component in 100 g of raw product—tabular or literature value (mg).

### 2.2.4. Assessing the Daily Requirement

The method is based on the calculation of the coverage of the daily demand for energy, nutrients, and pro-health based on nutrition standards and the calculation of the *INQ* index [44]. The calculations were made in duplicate.

### 2.2.5. Coverage of the Daily Energy Requirement

The calculations were adopted for adult women weighing 60 kg and for adult men weighing 70 kg, whose energy requirements are 2300 kcal and 3000 kcal, respectively.

$$CED = \frac{E}{SED} \times 100 , \tag{4}$$

where:

*CED*—coverage of daily energy demand (%);
*E*—energy content in 100 g of product (kcal);
*SE*—tabular value of standard energy demand (kcal).

### 2.2.6. Covering the Daily Nutrient Requirements

The calculations were adopted for adult women weighing 60 kg (fats—64 g; protein—48 g; carbohydrates—100 g) and for adult men weighing 70 kg (fats—82 g; protein—56 g; carbohydrates—100 g).

$$CDR = \frac{M_{so}}{S_{so}} \times 100, \tag{5}$$

where:

*CDR*—coverage of the daily requirement for a given nutrient (%);
$M_{SO}$—the content of a given nutrient in 100 g of the product (g);
$S_{SO}$—tabular value of the standard requirement for a given nutrient (g).

### 2.2.7. Covering the Daily Requirement for Health-Promoting Ingredients

Calculations were made for women on the basis of the recommended intake (vitamin A—0.7 mg; vitamin C—75 mg; niacin—14 mg; folates—0.4 mg, magnesium—310 mg; iron—13 mg). For vitamin E (8 mg), calcium (1000 mg), and potassium (4700 mg) based on sufficient intake. For men, these values were: vitamin A—0.9 mg; vitamin C—90 mg; niacin—16 mg; folates—0.4 mg, magnesium—400 mg, iron—10 mg, vitamin E—10 mg, calcium—1000 mg, potassium—4700 mg.

$$CDRC = \frac{M_{spc}}{S_{spc}} \times 100, \tag{6}$$

where:

*CDRC*—coverage of the daily requirement for a given pro-health component (%);
$M_{spc}$—the content of a given pro-health component in 100 g of the product (mg);
$S_{spc}$—tabular value of the standard demand for a given pro-health component (mg).

### 2.2.8. Nutritional Quality Index *INQ* (Index Nutritional Quality)

The *INQ* index was calculated for individual nutrients using the following formula:

$$INQ = \frac{IC \cdot SED}{E \cdot SDC} \times 100, \tag{7}$$

where:

*IC*—ingredient content in 100 g of product (g);
*SED*—the standard of energy demand (kcal);

*E*—energy content in 100 g of product (kcal);

*SDC*—standard of demand for a given component (g).

The *INQ* indicator shows the extent to which a given product meets the demand for a given nutrient in relation to the energy supplied.

Ingredient values close to 1 represent well-balanced products, and values greater than or lower than 1 represent products that contain the ingredient in excess or insufficient in relation to energy.

### 2.2.9. Converting the Indicators into Dried Products

The method was based on tabular data of water content in individual vegetables included in the product. According to Ciurzyńska et al. [4], dry matter content in the tested material was close to 100%, so for the purpose of this study, it was assumed that the products did not contain any water and totally consisted of dry matter, containing all of the nutrients.

$$M_{IC} = \left( \frac{M_{CC1}}{M_{d.m.1}} + \frac{M_{CC2}}{M_{d.m.2}} + \ldots + \frac{M_{CCn}}{M_{d.m.n}} \right) \cdot 100, \tag{8}$$

where:

$M_{IC}$—ingredient content in 100 g of dried product (g);

$M_{CC1}$, $M_{CC2}$, ... $M_{CCn}$—content of a given ingredient in 1, 2 and $n$th vegetables, respectively, included in the product, in the amount used in production (g);

$M_{d.m.1}$, $M_{d.m.2}$, ... $M_{d.m.n}$—dry matter content in 1, 2 and $n$th vegetable, respectively, included in the product (g).

### 2.3. Statistical Methods

The results were statistically analysed in the Statistica 13 program (Statsoft Inc., Tulsa, OK, USA). The one-way analysis of variance (one-way ANOVA) was used and homogeneous groups were determined using the Tukey's test at the significance level of $p = 0.05$.

## 3. Results and Discussion

### 3.1. Energy Value of Freeze-Dried Vegetable-Based Products

The energy value is strongly correlated with the nutrient content. For this reason, the concept of nutritional value is commonly referred to as energy content and essential nutrients. More precisely, the nutritional value determines the suitability of food products to meet the body's metabolic needs. The degree of nutrient utilisation is influenced by their balance, digestibility, and bioavailability. It is plausible that products with a diversified composition and containing various nutrients have a higher nutritional value [42–45]. There is also a relationship: the more the product contains more nutrients and the more these nutrients are absorbed by the human body, the greater the nutritional value of the product [44–47].

Figure 1 shows the average energy value of freeze-dried plant products produced with the use of vegetables and hydrocolloids. The values are presented in terms of 10 g of finished products, which is one snack portion. The energy value of the individual flavour variants was quite even and ranged from 14.5–16.5 kcal/10 g portion of the finished product. At the same time, no significant differences were found, regardless of the reformulation of the composition and the hydrocolloid used for producing snack bars. All products belong to the same homogeneous group. Bars: 3—CPCB$_A$ and 4—CPCB$_G$ had the highest energy value: 15.990 and 16.150 kcal/10 g, respectively. On the other hand, bars: 1 and 2 were characterised by the lowest energy value of 14.703 and 14.863 kcal/10 g, respectively.

The differences in the energy value for individual bar compositions are determined by the content of individual vegetables, the highest of which, according to the Regulation (EU) No. 1169/2011 of the European Parliament and of the Council of 25 October 2011 [42], are characteristic of corn and potatoes. The content of those ingredients in the bars obtained, irrespective of the hydrocolloids used, amounts to about 20% in total in samples 1 and 2,

around 60% in samples 3 and 4, and around 48% in samples 5 and 6, which results in a different energy value of the finished products (Table 1; Figure 1).

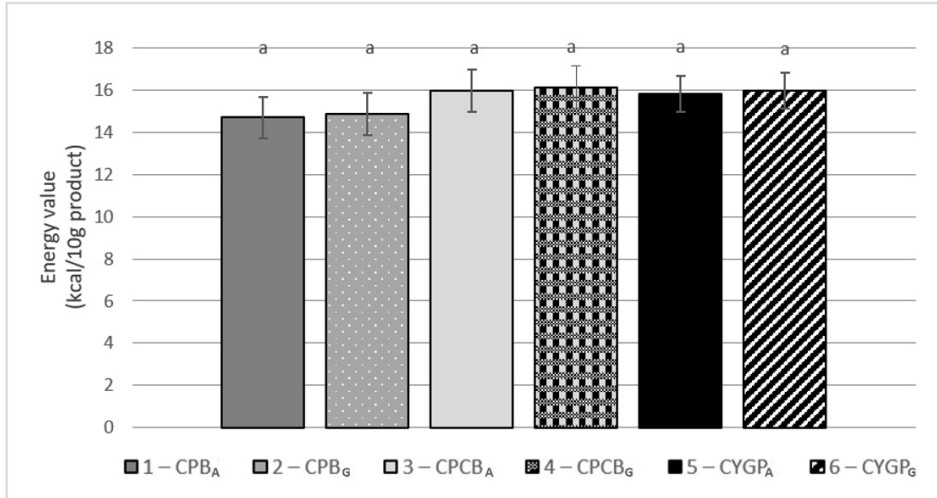

**Figure 1.** Average energy value of freeze-dried vegetable-based products produced with hydrocolloids. The same letters (a) in individual columns indicate the absence of a statistically significant difference between the samples (significance level 0.05). Table 1. Index (A) next to the formulations symbol—formulations with sodium alginate and calcium lactate, index (G) next to the formulations symbol—formulations with the mixture of locust bean gum and xanthan gum.

Manufactured by Gondek et al. [31], extruded cereal and vegetable snacks, including cereal–pumpkin and cereal–zucchini–broccoli, had an average energy value of 22.8 kcal per 10 g of product. Those values were slightly higher than those obtained in this study for presented bars with hydrocolloids.

In comparison, freeze-dried bars available in the market of the "Frupp" brand, made from fruit, have a higher energy value than the investigated vegetable bars. For example, a bar of apple (36 kcal), passion fruit and apple (34 kcal) and product to which a vegetable was added (banana, carrot, passion fruit) (37 kcal) [48]. Comparable energy value is also seen in fruit and nuts made by the "Farmer" company, which offer the flavour of spinach with lemon (36.7 kcal/10 g), tomato (35 kcal/10 g), and beetroot with garlic (34 kcal) [49]. "Foods by Ann" with dried strawberry and beetroot, presented as an ideal snack before exercise, and its energy value is 42.71 kcal per 10 g, is characterised by a higher calorific value among market vegetable bars [50].

### 3.2. Nutritional Value of Freeze-Dried Vegetable-Based Products

Nutrients are substances that may be absorbed by the body and used for energy, building or regulatory purposes. Among them there are essential ingredients, i.e., those that cannot be synthesised in the body and must be supplied with food. The most important nutrients are proteins, carbohydrates, and fats [51–53].

However, the nutrient content of a product is influenced by its composition, method of storage, transport, and processing. Every pre-treatment, such as washing, peeling and cutting vegetables, blanching, and freezing may result in the loss of nutrients, which was proven by Olivera et al. [54] for Brussels sprouts and Jaworska et al. [55] for mushrooms.

Figure 2a shows the nutritional value of freeze-dried vegetable-based products with the addition of hydrocolloids in the form of protein content. The highest protein value was found in bars 3—CPCB$_A$ and 4—CPCB$_G$—0.825 g/10g. The protein content in bars 1—CPB$_A$ and 2—CPB$_G$, as well as bars 5—CYGP$_A$ and 6—CYGP$_G$ is at an even level and amounts to 0.708 g/10g and 0.711g/10 g, respectively. Average protein content in bars is 0.75 g/10 g. The statistical analysis has shown no significant differences in the protein content of these products.

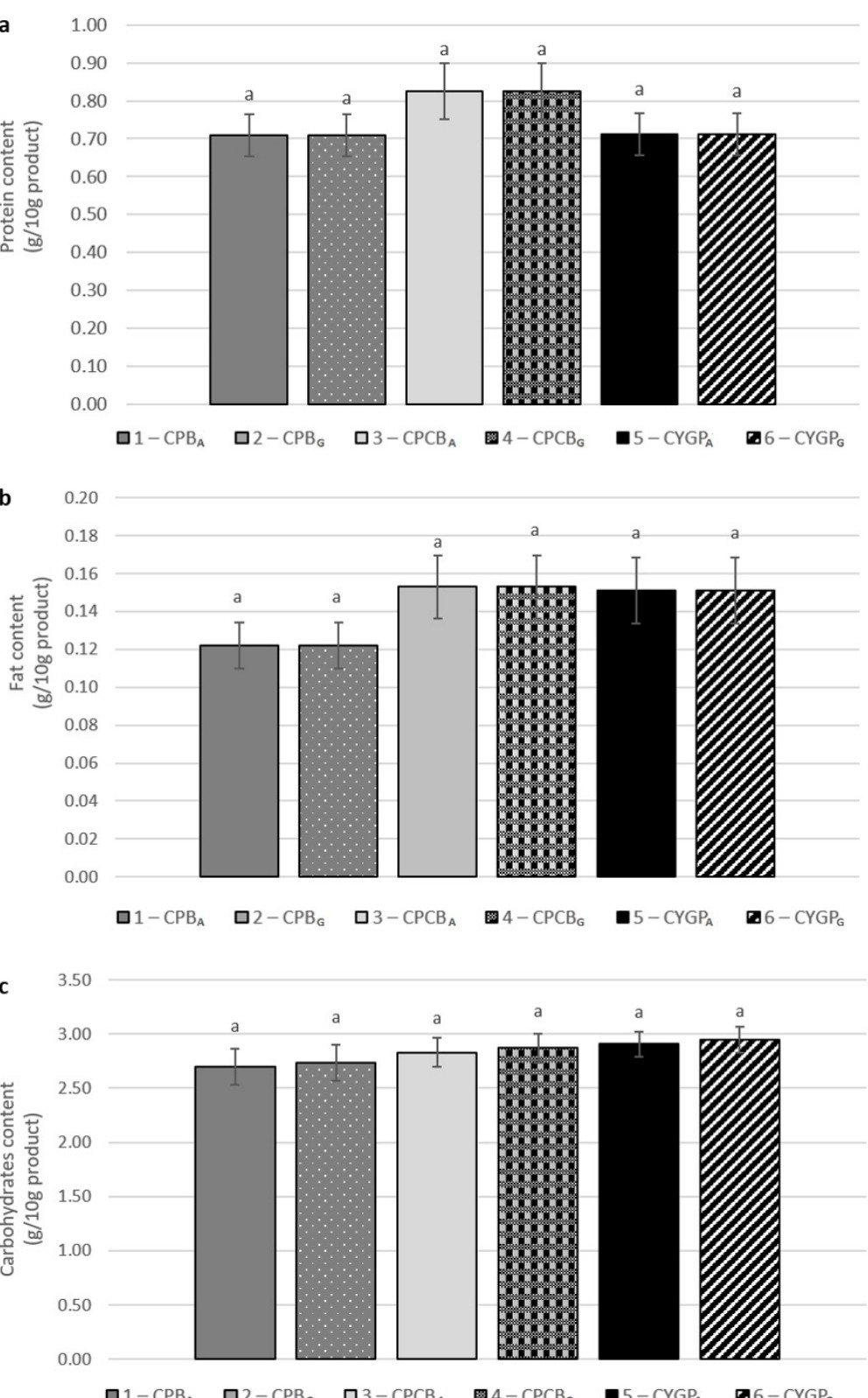

**Figure 2.** Average nutritional value of freeze-dried vegetable-based products produced with hydrocolloids: (**a**) protein; (**b**) fat; and (**c**) carbohydrates. The same letters (a) in individual columns indicate the absence of a statistically significant difference between the samples (significance level 0.05). Designations displayed in Table 1. Index (A) next to the formulations symbol—formulations with sodium alginate and calcium lactate, index (G) next to the formulations symbol—formulations with the mixture of locust bean gum and xanthan gum.

Proteins are among the most important nutrients because they have many functions. They account for a building component of cells, tissues, and body fluids of the human body and, in the form of enzymes, hormones, and antibodies, regulate metabolic processes. In human nutrition, they are responsible for stimulating the appetite and imparting aroma and flavour [51–53]. Proteins also constitute a source of amino acids that are used for synthesising body proteins. Some amino acids may be synthesised by the human body (endogenous amino acids), and others must be supplied with food (essential amino acids). For this reason, when evaluating food products in terms of protein, not only its quantity is important, but also the nutritional value of proteins, which is influenced by the content of individual amino acids and their mutual proportions [47–56]. Traditional, high-calorie snacks, such as crisps are characterized by a low protein content, on average 5–6 g/100 g [57]. In order to produce a high-protein product, an addition or combination of grains, legumes, or vegetables is used. Such research was carried out by Rababah et al. [58], who produced corn crisps with broad bean flour, chickpea flour, or isolated soy protein. It was shown that all crisps with the addition of flour had a higher nutritional value than traditional crisps. Protein contents increased from 6.4% to 10.5%. The others physical indicators were also improved. Fortification of food products with components, such as legume flours, is good solution to enhance the nutritional values of products. It has also been proven that the addition of a vegetable and herbal mixture resulted in an increase in nutrients. The addition of soybean ingredients to food products, such as cereal bars allow to obtain product with high isoflavone and soy protein content for use in diets to control dyslipidaemia [59]. The bars had only 245.47 kcal/100 g due to the high amount of dietary fibre 39.88 g/100 g, 34.25 g/100 g protein, and 100.39 mg/100 g isoflavones.

Carbohydrates, otherwise sugars or saccharides, constitute still another important nutrient that plays a significant role in the human body. They are made of carbon, hydrogen, and oxygen. Carbohydrates constitute the most easily digestible source of energy for the body's tissues. Moreover, they are the only energy material (glucose) for the nervous system and erythrocytes, because these cells are unable to use the energy derived from proteins or fats. Sugars are also used in the body as a reserve material in the muscles and liver (in the form of glycogen), they take part in the burning of fat from food, conditioning its proper metabolism and can participate in the synthesis of amino acids. In terms of human nutrition, an important role of saccharides is the regulation of the sense of hunger and satiety. In food technology, on the other hand, sugars determine the organoleptic features: they give flavour, participate in the creation of colour and ensure the structure of the product [60].

Figure 2b shows the fat content as one of the nutrients of the freeze-dried vegetable bars. The bar of 3—$CPCB_A$ and 4—$CPCB_G$—0.153 g/10 g and the bar of 5—$CYGP_A$ and 6—$CYGP_G$—0.151 g/10 g have the highest lipid content. Even for the 1—$CPB_A$ and 2—$CPB_G$ bars which obtained the lowest fat content (0.122 g/10 g) differences were insignificant from the other products. Average fat content in bars with hydrocolloids is 0.14 g/10 g.

Candy-type snacks, such as biscuits, chocolate, and chocolate bars contain a large amount of carbohydrates, but these are sugars added intentionally during the formulation stage of these products. On the other hand, snacks made of cereals, vegetables, fruit, or legume seeds provide natural sugars, and it is these raw materials that are the main source of sugar in the human diet. The carbohydrate content of legumes is generally around 60 g for beans, peas, and lentils. The exception is soybean, which contains about 32 g. Among vegetables, potatoes (20 g), and corn (23.4 g) have a high sugar content. The lowest carbohydrate content is found in leaf vegetables, such as lettuce (2.9 g) or spinach (3.0) and in those with a water content above 95%, i.e., tomato (3.6 g) and cucumber (2.9 g) [41–43].

The basic nutrient for the human body are fats or lipids. Fats constitute a good source of energy, providing 9 kcal g. Moreover, they constitute its backup form, depositing in adipose tissue, which also allows to maintain a constant body temperature and protect internal organs against injuries. Fats are also a source of essential unsaturated fatty acids

(EFA) and vitamins soluble in them (A, D, E, K). Taking into account the origin of fats, two groups can be distinguished: animal (butter, lard), which are the main source of cholesterol, and vegetable (oils, margarines). In the human diet, fats can come from food products (invisible fats) or they can be provided as separate foods, such as butter or vegetable oils (visible fats). Vegetable oils are obtained from seeds or fruits of oil plants, e.g., soybean oil or corn oil [41–43].

Figure 2c shows the carbohydrate content in bars produced with hydrocolloids, which is fairly even and ranges between 2.6 and 3.0 g/10 g. The bar 6—CYGP$_G$ (2.948 g/10 g) has the highest amount of sugars, and the bar 1—CPB$_A$ (2.698 g/10 g) has the lowest amount. Average carbohydrate content in bars with hydrocolloids is 2.83 g/10 g. Statistical analysis showed no significant differences for the carbohydrate content of these products.

Chips-type snacks, the traditional method of production of which is frying, are characterised by a high fat content of about 38–41 g/100 g [57]. Therefore, producers of crisps also use baking, which allows for the reduction in lipid content. An example is Lays oven-baked moulded potato chips, the paprika version of which contains 14 g fat/100 g [61]. For consumers looking for low-fat products, vegetable crisps may be an alternative, as vegetables belong to the group of products with a very low fat content. In most vegetables, the average fat content ranges from 0.2 to 0.5 g [42,43].

Gondek et al. [33], analysing the physical and functional properties of cereal and vegetable snacks produced by extrusion, obtained products with a lower protein and carbohydrate content, but with a comparable amount of fat than the tested bars without hydrocolloids. For example, the cereal and pumpkin snacks averaged 0.83 g protein, 5.44 g carbohydrates, and 0.22 g fat. On the other hand, cereal, zucchini, and broccoli snacks had 0.8 g of protein, 5.63 carbohydrates, and 0.17 g of fat. The significantly higher nutritional value of the tested, hydrocolloid-free freeze-dried vegetable-based products make them more attractive to a potential consumer. On the other hand, bars with the addition of hydrocolloids have a slightly lower content of nutrients than extruded snacks.

Comparing the tested vegetable-based products with the freeze-dried bars of the Frupp brand available on the market, it can be concluded that they are characterised by a higher nutritional value. Apple bar with passion fruit is characterised by a negligible amount of protein (0.09 g) and fat (0.02 g). A bar with carrots has a higher protein content—0.33 g, which may mean that the value of nutrients increases due to the addition of vegetables to the product. Only the content of carbohydrates in commercial products is close to their value in the bars tested and amounts to an average of 7.9 g [48].

Based on the examples provided, it can be seen that the nutritional value of snacks varies. It often results from the methods used in their production, which is not always beneficial to human health. Undoubtedly, however, the addition of plant products, mainly vegetables, allows to increase the nutritional value of the snacks produced.

### 3.3. Pro-Health Value of Freeze-Dried Vegetable-Based Products

In food products, apart from basic nutrients, there are also specific compounds or chemical substances having a health-promoting effect on the human body, protecting it against many diseases and ageing processes [42,43].

Figure 3 shows the average content of dietary fibre in freeze-dried products based on vegetables. The highest content of this component is found in bars 1—CPB$_A$ and 2—CPB$_G$—0.966 g/10 g. The remaining bars have an even fibre value with an average amount of 0.802 g/10 g. Statistical analysis showed no significantly different.

The health-promoting effect of vegetables is also due to the high content of dietary fibre (dietary fibre), which includes the edible parts of plants and some carbohydrates that are not digested in the small intestine, such as cellulose, lignin, or pectin [42,43]. Dietary fibre is responsible for regulating intestinal peristalsis and causes a quick feeling of fullness. Moreover, fibre inhibits the synthesis of cholesterol in the liver and reduce the prevalence of coronary heart diseases and cancer, as well as reduces diabetes and obesity risk [4,62]. Dietary fibre in vegetables occurs on average in the amount of 0.5–6%. Its

source is mainly cruciferous vegetables, such as white Brussels sprouts (3.8 g), kale (3.6 g), and root vegetables, i.e., carrots (3.6 g) and parsley (4.9 g). Komolka and Górecka [63], when assessing the effect of heat treatment of cruciferous vegetables on the content of dietary fibre, obtained the amount of this ingredient in fresh raw cabbage in the range from 12.66 g/100 g d.m for common to 15.48 g/100 g d.m for red. Moreover, they proved a significant influence of heat treatment: vegetables cooked in water had a higher content of dietary fibre than those cooked in steam [64,65].

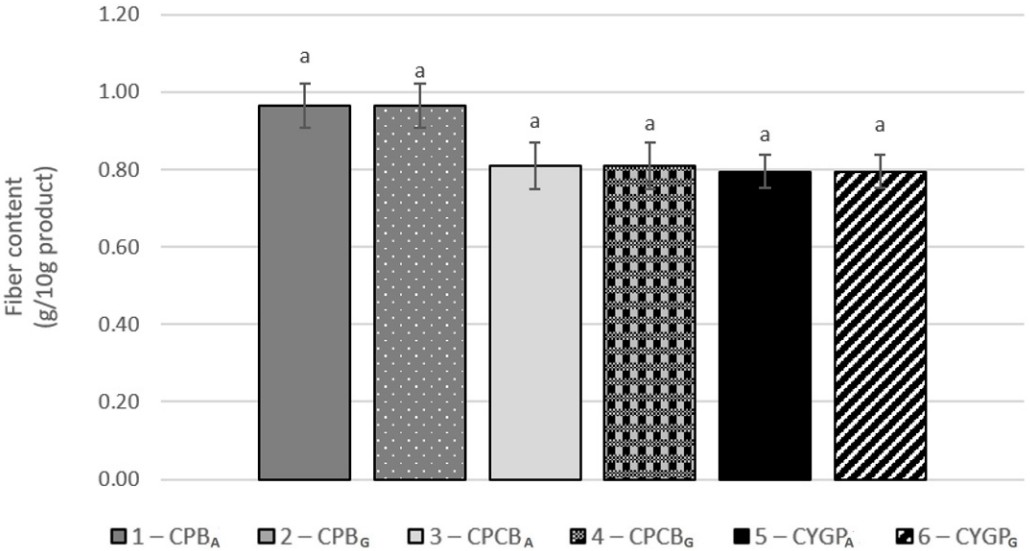

**Figure 3.** Average content of dietary fibre (pro-health value) in freeze-dried products formed on the basis of vegetables produced with hydrocolloids. The same letters (a) in individual columns indicate the absence of a statistically significant difference between the samples (significance level 0.05). Designations displayed in Table 1. Index (A) next to the formulations symbol—formulations with sodium alginate and calcium lactate, index (G) next to the formulations symbol—formulations with the mixture of locust bean gum and xanthan gum.

Gondek et al. [31], analysing the physical and functional properties of cereal and vegetable snacks produced by extrusion, obtained products with a lower content of dietary fibre. For example, cereal-pumpkin snacks contained an average of 1.52 g/10 g, and cereal–zucchini–broccoli snacks 0.80 g/10 g.

There are also vegetable bars with lower fibre content available on the market, such as for example the product with dried strawberry and beetroot "Foods by Ann", which contains 1.28 g/10 g of this health-promoting ingredient, or a "FreeYu" bar with carrot, apple, and orange (0.55 g/10 g) [50–66].

Snacks in the form of fried chips also contain certain amounts of vitamins (vitamin C, E, thiamine, riboflavin, niacin) and minerals (sodium, potassium, phosphorus, magnesium), but their pro-health effects are limited by high caloric content, high fat and sodium chloride content [57]. Another popular snack with a health-promoting effect can be muesli bars, due to the high content of dietary fibre in them. Sikora [67], assessing the nutritional value of bars available on the market, obtained an average of 7.49% of dietary fibre. In addition, it was proved that the addition of dried fruit, nuts or extruded grains increased the content of this ingredient in the tested bars. The dried fruit itself can also be treated as a snack, and their health-promoting value results mainly from the content of polyphenols and vitamins in them. Kovacev et al. [68] investigated the polyphenol content and stability in sweetened dried cranberries between product matrix types. They showed that sweetened dried cranberry polyphenols are unstable regardless of product matrix. The optimal processing parameters for sweetened dried cranberries to maintain polyphenol stability as healthier food options for consumers have to be investigated.

Table 2 shows the average content of vitamins in freeze-dried vegetable-based products with the addition of hydrocolloids.

**Table 2.** Average content of selected vitamins (pro-health value) in freeze-dried products formed on the basis of vegetables produced with hydrocolloids.

| Material | Layer | Vit. C | Vit. E | Vit. A | Niacin | Foils | β-Carotene |
|---|---|---|---|---|---|---|---|
| | | (mg/10 g Product) | | | | | |
| 1—CPB$_A$<br>2—CPB$_G$ | I | 0.671 ± 0.215 | 0.076 ± 0.047 | 0.094 ± 0.005 | 0.112 ± 0.040 | 0.002 ± 0.001 | 0.940 ± 0.222 |
| | II | 4.658 ± 0.667 | 0.007 ± 0.000 | 0.000 ± 0.000 | 0.077 ± 0.007 | 0.005 ± 0.001 | 0.000 ± 0.000 |
| | III | 10.677 ± 5.943 | 0.245 ± 0.085 | 0.023 ± 0.006 | 0.180 ± 0.048 | 0.011 ± 0.002 | 0.258 ± 0.046 |
| | Overall | 16.006 ± 6.825 [a] | 0.328 ± 0.132 [a] | 0.117 ± 0.011 [b] | 0.369 ± 0.095 [a] | 0.018 ± 0.004 [a] | 1.198 ± 0.268 [b] |
| 3—CPCB$_A$<br>4—CPCB$_G$ | I | 0.597 ± 0.073 | 0.003 ± 0.000 | 0.000 ± 0.000 | 0.087 ± 0.005 | 0.002 ± 0.000 | 0.000 ± 0.000 |
| | II | 4.645 ± 0.656 | 0.010 ± 0.001 | 0.001 ± 0.000 | 0.079 ± 0.008 | 0.005 ± 0.001 | 0.014 ± 0.001 |
| | III | 10.001 ± 5.584 | 0.284 ± 0.014 | 0.025 ± 0.008 | 0.201 ± 0.058 | 0.012 ± 0.003 | 0.288 ± 0.059 |
| | Overall | 15.243 ± 6.313 [a] | 0.297 ± 0.014 [a] | 0.026 ± 0.008 [a] | 0.367 ± 0.070 [a] | 0.020 ± 0.004 [a] | 0.301 ± 0.060 [a] |
| 5—CYGP$_A$<br>6—CYGP$_G$ | I | 0.934 ± 0.147 | 0.021 ± 0.002 | 0.000 ± 0.000 | 0.096 ± 0.005 | 0.003 ± 0.000 | 0.000 ± 0.000 |
| | II | 15.184 ± 7.901 | 0.175 ± 0.014 | 0.029 ± 0.003 | 0.145 ± 0.036 | 0.010 ± 0.003 | 0.358 ± 0.057 |
| | III | 4.618 ± 0.661 | 0.007 ± 0.000 | 0.000 ± 0.000 | 0.076 ± 0.007 | 0.005 ± 0.001 | 0.000 ± 0.000 |
| | Overall | 20.736 ± 8.708 [a] | 0.202 ± 0.017 [a] | 0.029 ± 0.003 [a] | 0.317 ± 0.047 [a] | 0.018 ± 0.004 [a] | 0.358 ± 0.057 [a] |

[a,b]—same letters in the column indicate no significant difference at $p = 0.05$.

The highest content of vitamin C is found in bars 5—CYGP$_A$ and 6—CYGP$_G$ (20.736 mg/10 g), which is mainly influenced by their second layer, because its main component is paprika. The lowest content of this vitamin is found in bars 3—CPCB$_A$ and 4—CPCB$_G$—15.243 mg/10 g. The average amount of vitamin C in hydrocolloid bars is 17.33 mg/10 g. Statistical analysis showed no significant differences between the bars.

The highest content of vitamin E is found in bars 1—CPB$_A$ and 2—CPB$_G$ (0.328 mg/10 g) and it is mainly affected by the third layer, as about 75% of this ingredient comes from this layer. The lowest content of this vitamin is found in bars 5—CYGP$_A$ and 6—CYGP$_G$—0.202 mg/10 g. The average vitamin E content in bars with hydrocolloids is 0.28 mg/10 g. The statistical analysis has shown no significant differences between the bars.

The highest and significantly different vitamin A content was found in bars 1—CPB$_A$ and 2—CPB$_G$—0.117 mg/10 g. In the remaining bars with hydrocolloids, the amount of this ingredient is at a significantly lower and even level and averages 0.028 mg/10 g.

The content of niacin is at a fairly even level, which was confirmed by the statistical analysis, which showed no significant differences between the bars. The largest amount of this ingredient is contained in bars 1—CPB$_A$ and 2—CPB$_G$—0.369 mg/10 g. The average niacin content in bars with hydrocolloids is 0.35 mg/10 g.

The tested bars are characterised by a fairly trace amount of folate. The highest content of this ingredient is found in bars of 3—CPCB$_A$ and 4—CPCB$_G$—0.020 mg, while the average amount of this ingredient for all products with hydrocolloids is 0.019 mg/10 g. The statistical analysis has not shown any significant differences between the bars in terms of their folate content.

The highest and significantly different content of β-carotene is found in bars 1—CPB$_A$ and 2—CPB$_G$—1.198 mg/10 g, which is mainly influenced by their first layer, because its main component is carrot. The amount of beta-carotene in the remaining hydrocolloid bars is at an even level and amounts to an average of 0.330 mg/10 g.

The Crispy Natural Dried Vegetable Chips available in the market have a higher vitamin content. For example, carrot chips with spices (including dried peppers) contain 0.17 mg/10 g of vitamin A. This value is only close to the 1—CPB$_A$ and 2—CPB$_G$ bars. Beet crisps, on the other hand, contain a similar, but greater amount of folate (0.05 mg/10 g) than the content of this ingredient in bars without hydrocolloids. This may mean that in order to increase the folate content in the tested bars, which is negligible, beet should be used as one of their ingredients [69].

Compared to fruit, vegetables are characterised by an even higher pro-health value, which is why they are more and more often used in the production of snacks.

The pro-health value of vegetables is primarily determined by the high content of vitamins, i.e., organic compounds that are necessary for proper functioning and maintaining health. They are not synthesised in the body, therefore they must be supplied with food. Vegetables are characterised by a high content of vitamin C (ascorbic acid), which has antioxidant properties and is responsible for the biosynthesis of collagen and carnitine. In addition, it normalises blood pressure, supports the absorption of iron and calcium, and strengthens the functioning of the immune system [70]. The greatest amounts of vitamin C can be found in cruciferous vegetables, such as broccoli (93.2 mg) or cauliflower (48.2 mg) [71]. Moreover, red pepper (127 mg) and leaf parsley (133 mg) have a high content of ascorbic acid. Additionally, vegetables are a rich source of vitamin A (retinol and its derivatives), which is involved in the process of vision, tissue growth and is responsible for maintaining the proper condition of the skin. The highest amounts of this vitamin are found in vegetables, such as carrot (1656 μg), kale (892 μg), pumpkin (496 μg), and broccoli (153 μg) [42,43]. Vegetables also contain large amounts of vitamin E, which is involved in metabolic processes and is a strong antioxidant [72]. Additionally, vegetables have B vitamins, niacin, folic acid, and some of them also vitamin K.

The health-promoting value of vegetables is also due to the fact that they constitute a valuable source of minerals, i.e., elements that remain after tissue mineralization in the form of ash. In vegetables, the predominant iron content is responsible for the transport of oxygen and its storage in the body. Moreover, this element provides immunity [73]. The greatest amounts of iron are found primarily in leaf vegetables, such as spinach (2.8 mg) and parsley leaves (5.3 mg) [43]. Vegetables are also characterised by a high content of calcium, which is a building component of bones, teeth, hair, and nails, and plays a role in blood clotting and conduction of impulses. The greatest amounts of this element are found in cruciferous vegetables, which on average contain about 48 mg of this ingredient. The highest value among this group is achieved by kale (157 mg) [74]. Potassium is also an important mineral component of vegetables, which, together with sodium and chlorine, regulate the water balance and influence the acid-base balance [75]. Vegetables such as tomato (282 mg), potato (491 mg), or celery (320 mg) have a high potassium content [43]. The health-promoting value of vegetables can also be considered in terms of their antioxidant properties. Such substances are able to inhibit the reaction with oxygen and ozone and neutralise the action of free radicals that contribute to oxidative stress, causing diseases, such as cancer, stroke, or heart attack. Antioxidant compounds are primarily vitamin C, tocopherols, carotenoids, betalains, chlorophylls, and polyphenols [76].

Carotenoids constitute a group of lipophilic compounds that give the colour from yellow to red. Their antioxidant activity is mainly related to the quenching of singlet oxygen [77]. Carotenoids include substances, such as β-carotene, lycopene, lutein, and zeaxanthin. The main source of β-carotene among vegetables is carrot (9.02 mg/100 g), kale (7.28 mg/100 g), parsley (5.5 mg/100 g), red bell pepper (3.25 mg/100 g), pumpkin (0.2 mg/100 g), and broccoli (0.28 mg/100 g).

Table 3 presents the average content of selected minerals in the freeze-dried products formed on the basis of vegetables produced with hydrocolloids.

The iron content is at a fairly even level, which is confirmed by the statistical analysis, which has shown no significant differences between the bars. The highest content of this ingredient is distinguished by bars 3—$CPCB_A$ and 4—$CPCB_G$ (0.389 mg/10 g), and the lowest by bars 5—$CYGP_A$ and 6—$CYGP_G$ (0.317 mg/10 g). Average iron content in bars with hydrocolloids is 0.34 mg/10 g.

The highest calcium content is found in bars 1—$CPB_A$ and 2—$CPB_G$ (16.533 mg/10 g), which is mainly influenced by their third layer. The smallest amount of this mineral is found in bars 5—$CYGP_A$ and 6—$CYGP_G$—13.327 mg/10 g. The average calcium content for all products with hydrocolloids is 15.46 mg/10 g. Statistical analysis showed no significant differences between the bars.

Among the selected minerals, the potassium content is at the highest level. The highest amount of this mineral is found in bars 1—$CPB_A$ and 2—$CPB_G$ (110.012 mg/10 g) and it

is levelled for all their layers. Bars 3—CPCB$_A$ and 4—CPCB$_G$ have the lowest content—93.048 mg/10 g. The average amount of this ingredient for all hydrocolloid products is 101.43 mg/10 g. Bars 1—CPB$_A$ and 2—CPB$_G$ 2 differ significantly from the others in terms of potassium content.

**Table 3.** Average content of selected minerals (pro-health value) in freeze-dried products formed on the basis of vegetables produced with hydrocolloids.

| Material | Layer | Iron | Calcium | Potassium | Magnesium |
|---|---|---|---|---|---|
| | | (mg/10g Product) | | | |
| 1—CPB$_A$ 2—CPB$_G$ | I | 0.034 ± 0.003 | 3.733 ± 0.374 | 36.156 ± 3.220 | 1.356 ± 0.130 |
| | II | 0.061 ± 0.007 | 2.223 ± 0.276 | 38.602 ± 3.306 | 1.990 ± 0.150 |
| | III | 0.232 ± 0.073 | 10.576 ± 2.823 | 35.253 ± 8.138 | 3.289 ± 0.839 |
| | **Overall** | **0.328 ± 0.083** [a] | **16.533 ± 3.474** [a] | **110.012 ± 14.664** [a] | **6.635 ± 1.118** [a] |
| 3—CPCB$_A$ 4—CPCB$_G$ | I | 0.034 ± 0.004 | 0.280 ± 0.105 | 17.939 ± 1.367 | 1.893 ± 0.203 |
| | II | 0.070 ± 0.013 | 2.749 ± 0.532 | 38.997 ± 3.367 | 2.038 ± 0.172 |
| | III | 0.284 ± 0.096 | 13.478 ± 3.641 | 36.113 ± 10.493 | 3.501 ± 1.083 |
| | **Overall** | **0.389 ± 0.112** [a] | **16.507 ± 4.277** [a] | **93.048 ± 15.227** [a] | **7.431 ± 1.459** [a] |
| 5—CYGP$_A$ 6—CYGP$_G$ | I | 0.063 ± 0.009 | 1.605 ± 0.205 | 18.959 ± 1.429 | 2.450 ± 0.247 |
| | II | 0.194 ± 0.068 | 9.515 ± 4.143 | 43.916 ± 7.564 | 3.707 ± 0.996 |
| | III | 0.061 ± 0.007 | 2.206 ± 0.275 | 38.351 ± 3.283 | 1.978 ± 0.149 |
| | **Overall** | **0.317 ± 0.084** [a] | **13.327 ± 4.623** [a] | **101.227 ± 12.276** [a] | **8.134 ± 1.392** [a] |

[a]—same letters in the column indicate no significant difference at *p* = 0.05.

The highest magnesium content is found in bars 5—CYGP$_A$ and 6—CYGP$_G$—8.134 mg/10 g, while bars 1—CPB$_A$ and 2—CPB$_G$ 6.635 mg/10 g are the lowest. The average amount of this ingredient for all hydrocolloid products is 7.40 mg/10 g. Statistical analysis showed that bars 3, 4, 5, and 6 belong to one homogeneous group and differ significantly from bars 1 and 2.

Crispy natural dried vegetable chips available on the market have a varied potassium content. For example, carrots (180 mg) and sweet potatoes (210 mg) have a lower amount of potassium than the bars without hydrocolloids tested in this study. On the other hand, the potassium content in the tomato product (351 mg) significantly exceeds the amounts obtained for the tested bars.

Taking into account the above changes during food processing, the starting product for creating snacks with a high content of pro-health ingredients should be raw materials with high nutritional value. As presented above, vegetables are undoubtedly such a raw material. Their low energy value and high content of bioactive ingredients prove that the share of vegetables in nutrition is not related to satisfying energy, but to health-promoting properties and taste. For this reason, vegetables are more and more often used as an addition to snacks or as their main ingredient, but to maintain the highest health-promoting value, the method of their processing should be properly selected [31,78].

### 3.4. Assessment of Nutritional Quality of Freeze-Dried Vegetable-Based Products

Table 4 shows the average coverage of the daily requirement for nutrients (protein, fat, carbohydrates) and energy in the human diet by freeze-dried products based on vegetables produced with hydrocolloids.

Daily protein requirements are best covered by bars of 3—CPCB$_A$ and 4—CPCB$_G$. This coverage is 1.72% for women and 1.47% for men, which means that to cover 100% of the demand would have to eat about 58 and 68 bars, respectively. The average daily protein coverage for all hydrocolloid bars is 1.45%. Coverage of the daily fat requirement is very low. The average for all bars with hydrocolloids is 0.2%, which means that to cover 100% of the fat requirement, would have to eat as many as 500 bars. The highest coverage of the

daily requirement was obtained for carbohydrates, which was in the range of 2.7–3.0%. Its highest value was achieved by bars 5—CYGP$_A$ and 6—CYGP$_G$—2.95%, which meant that to cover 100% of the demand for sugar, about 34 bars should be consumed. The average coverage of the daily energy requirement was quite low. They were best covered by the 4—CPCB$_G$ bar—0.70% for women and 0.54% for men. Average daily energy coverage for all hydrocolloid bars was 0.60%.

**Table 4.** Average coverage of the daily requirement (CDR) for protein, fat, carbohydrates, and energy in the human diet by freeze-dried products based on vegetables produced with hydrocolloids (10 g).

| Material | Gender | Coverage of the Daily Requirement (%) | | | |
|---|---|---|---|---|---|
| | | Protein | Fat | Carbohydrates | Energy |
| 1—CPB$_A$ | Woman | 1.48 ± 0.23 [B] | 0.19 ± 0.04[b] | 2.70 ± 0.33 [1] | 0.64 ± 0.09 [A,B] |
| | Man | 1.26 ± 0.20 [C] | 0.15 ± 0.03[c] | 2.70 ± 0.33 [1] | 0.49 ± 0.07 [C] |
| 2—CPB$_G$ | Woman | 1.48 ± 0.23 [B] | 0.19 ± 0.04 [b] | 2.74 ± 0.33 [1] | 0.65 ± 0.09 [B] |
| | Man | 1.60 ± 0.20 [A,B] | 0.15 ± 0.03 [c] | 2.74 ± 0.33 [1] | 0.50 ± 0.07 [C] |
| 3—CPCB$_A$ | Woman | 1.72 ± 0.31 [A] | 0.24 ± 0.05 [a] | 2.83 ± 0.26 [1] | 0.70 ± 0.08 [A] |
| | Man | 1.47 ± 0.27 [B] | 0.19 ± 0.04 [b] | 2.83 ± 0.26 [1] | 0.53 ± 0.06 [C] |
| 4—CPCB$_G$ | Woman | 1.72 ± 0.31 [A] | 0.24 ± 0.05 [a] | 2.87 ± 0.26 [1] | 0.70 ± 0.08 [A] |
| | Man | 1.47 ± 0.27 [B] | 0.19 ± 0.04 [b] | 2.87 ± 0.26 [1] | 0.54 ± 0.06 [C] |
| 5—CYGP$_A$ | Woman | 1.48 ± 0.23 [B] | 0.24 ± 0.06 [a] | 2.91 ± 0.24 [1] | 0.69 ± 0.07 [A] |
| | Man | 1.27 ± 0.20 [C] | 0.18 ± 0.04 [b] | 2.91 ± 0.24 [1] | 0.53 ± 0.06 [C] |
| 6—CYGP$_G$ | Woman | 1.48 ± 0.23 [B] | 0.24 ± 0.06 [a] | 2.95 ± 0.24 [1] | 0.69 ± 0.07 [A] |
| | Man | 1.27 ± 0.20 [C] | 0.18 ± 0.04 [b] | 2.95 ± 0.24 [1] | 0.53 ± 0.06 [C] |

[A–C, a–c, 1, A–C]—same letters or numbers in the column indicate no significant difference at *p* = 0.05 in terms of gender.

Coverage of the daily requirement can be compared with the Reference Daily Intake (RDI) shown on the labels of commercial products. By comparing the tested freeze-dried vegetable-based products with Frupp bars available on the market, it can be concluded that they meet the human body's needs for nutrients to a greater extent. For example, an apple bar has an RDI which is 3% for carbohydrates and 0% for fat and protein. A bar with a vegetable added—banana, carrot, apple has slightly higher values: 4% for carbohydrates, 1% for protein, and 0% fat. Coverage of energy demand is greater for commercial bars, amounting to 2% for both the mentioned products [48]. The Crispy natural beetroot vegetable chips available on the market have a reference intake of 2.2% for protein and carbohydrates and 0% for fat per 10 g of product. These values are also lower, comparing them to the daily requirement coverage for bars without hydrocolloids. Hydrocolloid bars, on the other hand, have a lower CDR value for protein. Commercial beet chips meet the energy needs to a slightly better extent, as the reference intake for this value is 1.7% [79].

Table 5 shows the average coverage of the daily requirement for selected vitamins in the human diet by freeze-dried products based on vegetables produced with hydrocolloids.

Coverage of the daily requirement for vitamin C has reached the highest levels. They are best satisfied by the bars of 5—CYGP$_A$ and 6—CYGP$_G$—27.65% for women and 23.04% for men, which means that in order to cover 100% of the daily requirement, it would be necessary to consume 4 and 5 bars, respectively. The average coverage of the daily vitamin C requirement for all hydrocolloid bars is 21.18%.

The highest daily coverage of vitamin E requirements is found in bars 1—CPB$_A$ and 2—CPB$_G$, for which this value is 4.10% for women and 3.28% for men. The average value of this indicator for all bars with hydrocolloids is 3.10%.

The *INQ* indicator shows the extent to which a given product meets the demand for a given nutrient in relation to the energy supplied. Indicators close to 1 represent

well-balanced products, and values greater than or lower than 1 represent products that contain the ingredient in excess or insufficient in relation to energy.

**Table 5.** Average coverage of the daily requirement (CDR) for selected vitamins in the human diet by freeze-dried products based on vegetables produced with hydrocolloids (10 g).

| Material | Gender | Coverage of the Daily Requirement (%) | | | | |
|---|---|---|---|---|---|---|
| | | Vit. C | Vit. E | Vit. A | Niacin | Foils |
| 1—CPB$_A$ | Woman | 21.34 ± 9.10 [A] | 4.10 ± 1.65 [a] | 16.76 ± 1.59 [1] | 2.64 ± 0.68 [A] | 4.60 ± 1.01 [a,b] |
| | Man | 17.78 ± 7.58 [A,B] | 3.28 ± 1.32 [b,c] | 13.03 ± 1.24 [2] | 2.31 ± 0.59 [A,B] | 4.60 ± 1.01 [a,b] |
| 2—CPB$_G$ | Woman | 21.34 ± 9.10 [A] | 4.10 ± 1.65 [a] | 16.76 ± 1.59 [1] | 2.64 ± 0.68 [A] | 4.60 ± 1.01 [a,b] |
| | Man | 17.78 ± 7.58 [A,B] | 3.28 ± 1.32 [b,c] | 13.03 ± 1.24 [2] | 2.31 ± 0.59 [A,B] | 4.60 ± 1.01 [a,b] |
| 3—CPCB$_A$ | Woman | 20.32 ± 8.42 [A] | 3.71 ± 1.43 [a,b] | 3.74 ± 1.17 [2] | 2.62 ± 0.50 [A] | 4.93 ± 1.06 [a] |
| | Man | 16.94 ± 7.01 [A,B] | 2.97 ± 1.14 [b,c] | 2.91 ± 0.91 [3,4] | 2.29 ± 0.44 [A,B] | 4.93 ± 1.06 [a] |
| 4—CPCB$_G$ | Woman | 20.32 ± 8.42 [A] | 3.71 ± 1.43 [a,b] | 3.74 ± 1.17 [2] | 2.62 ± 0.50 [A] | 4.93 ± 1.06 [a] |
| | Man | 16.94 ± 7.01 [A,B] | 2.97 ± 1.14 [b,c] | 2.91 ± 0.91 [3,4] | 2.29 ± 0.44 [A,B] | 4.93 ± 1.06 [a] |
| 5—CYGP$_A$ | Woman | 27.65 ± 11.61 [A] | 2.53 ± 0.21 [c,d] | 4.21 ± 0.44 [2] | 2.26 ± 0.34 [A,B] | 4.60 ± 0.98 [a,b] |
| | Man | 23.04 ± 9.68 [A] | 2.02 ± 0.17 [d] | 3.27 ± 0.34 [2,3] | 1.98 ± 0.29 [B,C] | 4.60 ± 0.98 [a,b] |
| 6—CYGP$_G$ | Woman | 27.65 ± 11.61 [A] | 2.53 ± 0.21 [c,d] | 4.21 ± 0.44 [2] | 2.26 ± 0.34 [A,B] | 4.60 ± 0.98 [a,b] |
| | Man | 23.04 ± 9.68 [A] | 2.02 ± 0.17 [d] | 3.27 ± 0.34 [2,3] | 1.98 ± 0.29 [B,C] | 4.60 ± 0.98 [a,b] |

[A,B, a–d, 1–4, A–C, a,b]—same letters or numbers in the column indicate no significant difference at $p = 0.05$ in terms of gender.

The values of this index for protein are kept within the range of 2.10–2.80, which is far from the reference value 1, for which the products are well-balanced in relation to energy. Carbohydrates, for which the values range from 4 to 6, have an even higher level of this index. Such a high *INQ* index means that protein and sugars are in excess in the tested bars compared to the energy they supply. The value of the nutritional quality index for fat is too low and amounts to 0.33 on average, which means that the lipid content in hydrocolloid bars is insufficient (Table 6).

**Table 6.** Average of the Nutritional Quality Index (*INQ*) for freeze-dried products formed on the basis of vegetables produced with hydrocolloids.

| Material | Gender | Nutritional Quality Index (*INQ*) | | |
|---|---|---|---|---|
| | | Protein | Fat | Carbohydrates |
| 1—CPB$_A$ | Woman | 1.48 ± 0.23 [B] | 0.19 ± 0.04 [b] | 2.70 ± 0.33 [1] |
| | Man | 1.26 ± 0.20 [C] | 0.15 ± 0.03 [c] | 2.70 ± 0.33 [1] |
| 2—CPB$_G$ | Woman | 1.48 ± 0.23 [B] | 0.19 ± 0.04 [b] | 2.74 ± 0.33 [1] |
| | Man | 1.60 ± 0.20 [A,B] | 0.15 ± 0.03 [c] | 2.74 ± 0.33 [1] |
| 3—CPCB$_A$ | Woman | 1.72 ± 0.31 [A] | 0.24 ± 0.05 [a] | 2.83 ± 0.26 [1] |
| | Man | 1.47 ± 0.27 [B] | 0.19 ± 0.04 [b] | 2.83 ± 0.26 [1] |
| 4—CPCB$_G$ | Woman | 1.72 ± 0.31 [A] | 0.24 ± 0.05 [a] | 2.87 ± 0.26 [1] |
| | Man | 1.47 ± 0.27 [B] | 0.19 ± 0.04 [b] | 2.87 ± 0.26 [1] |
| 5—CYGP$_A$ | Woman | 1.48 ± 0.23 [B] | 0.24 ± 0.06 [a] | 2.91 ± 0.24 [1] |
| | Man | 1.27 ± 0.20 [C] | 0.18 ± 0.04 [b] | 2.91 ± 0.24 [1] |
| 6—CYGP$_G$ | Woman | 1.48 ± 0.23 [B] | 0.24 ± 0.06 [a] | 2.95 ± 0.24 [1] |
| | Man | 1.27 ± 0.20 [C] | 0.18 ± 0.04 [b] | 2.95 ± 0.24 [1] |

[A–C, a–c, 1]—same letters or numbers in the column indicate no significant difference at $p = 0.05$ in terms of gender.

## 4. Conclusions

Nowadays, the popularity of snack foods are increasing due to the fast-paced lifestyle of society. Thanks to the prevailing trends related to a healthy lifestyle and organic food, the need to create new products is increasing, but also more and more attention is paid to high nutritional value. Freeze-drying is a drying method that allows in popularity to create a product with a retained structure, colour, high pro-health value, and good microbiological stability. By using this technique for drying vegetables and shaping them into a bar shape, an attractive and original snack for consumers can be obtained.

The studied features of the produced multi-layer vegetable bars allow to state that the introduction of this type of snacks into the diet is possible and satisfactory in terms of nutrition. However, these are not products that fully cover the dietary needs of the nutritional components of the diet. At the same time, it can be assumed that properly selected meal ingredients and the consumption of this type of snacks is beneficial for a potential consumer. At the same time, the obtained results open the way for further research in the field of reformulation of the composition of the proposed products in the aspect of developing new innovative products in the form of snacks that fit into the sustainable development strategy of the healthy functional snacks sector as an alternative to sweets.

According to results obtained due to presented research, freeze-dried multi-layer vegetable snacks with addition of hydrocolloids were characterized by low calories equal to 14.5–16.5 kcal per 10 g of the product, which portrayed one serving. Reformulation had no significant effect on the nutritional value of developed products. The main components of the snacks were carbohydrates and dietary fibre. Freeze-dried vegetable snacks were a rich source of vitamin C and minerals, especially calcium. Moreover, carrot contained in the formulation of the snacks caused an increase in the daily requirement coverage (CDR) for vitamin A. Because of the low energy value, the snacks turned out to be not well-balanced and Nutritional Quality Indexes (NQI) and their carbohydrates and protein contents were over the optimum, while fat supply was found insufficient.

On the base of nutritional characteristics of the developed multi-layer vegetable bars, it should be emphasize that introduction of this type of snacks into human diet may be an interesting and attractive substitute for traditional snacks. However, those snacks cannot be defined as products fully covering needs for nutrients on daily basis. At the same time, it can be assumed that consumption of this type of snacks as complementation of properly balanced diet may have beneficial effect on functioning of a human organism.

Conducted research opened the way for further investigation in the field of reformulation of the composition of products similar to those that are described in this paper. Findings of this research can be implemented in the aspect of developing innovative snack products that would fit in the category of healthy functional snacks, sweets alternatives, and be produced in line with sustainable development strategy.

**Author Contributions:** Conceptualisation, M.J. and A.C.; methodology, M.J.; software, M.J.; validation, M.J., A.C. and S.G.; formal analysis, M.J.; investigation, M.J. and A.C.; resources, M.J.; data curation, M.J.; writing—original draft preparation, M.J., A.C. and M.K.; writing—review and editing, M.J., A.C., M.K., J.K. and S.G.; visualisation, M.J. and A.C.; supervision, M.J., A.C., J.K. and S.G.; project administration, M.J.; funding acquisition, M.J. All authors have read and agreed to the published version of the manuscript.

**Funding:** This work was founded by the National Center for Research and Development, as part of the III BIOSTRATEG. "The development of an innovative carbon footprint calculation method for the basic basket of food products"—task in the project "Development of healthy food production technologies taking into consideration nutritious food waste management and carbon footprint calculation methodology" BIOSTRATEG3/343817/17/NCBR/2018 and was also co–finances by a statutory activity subsidy from Polish Ministry of Sciences and Higher Education for the Faculty of Food Sciences of Warsaw University of Life Sciences.

**Institutional Review Board Statement:** Not applicable.

**Informed Consent Statement:** Not applicable.

**Data Availability Statement:** The data presented in this study are available on request from the corresponding author.

**Acknowledgments:** The authors would like to thank Inga Liszewska for participation in the experimental part of the research.

**Conflicts of Interest:** The authors declare no conflict of interest.

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
