# Peer review of "Mathematical Estimation of the Energy, Nutritional and Health-Promoting Values of Multi-Layer Freeze-Dried Vegetable Snacks"

_applsci, doi:10.3390/app12136379_

Round 1

Reviewer 1 Report

The manuscript dealt with mathematical estimation of the energy, nutritional and health promoting values of multi-layer freeze-dried vegetable snacks. There are some questions needed to be addressed.

Abstract: Q: Please rewrite Abstract section. Abstract section is too descriptive. Please include appropriate information such as the methods used in the study and the important data and results obtained from the study.

Lines 228-242 and lines 276-282: Q: Why the energy calculated using different units?

Equation 4: Q: Please rewrite the equation using appropriate mathematical symbol.

Tables 4-6: Q: Please provide the statistical analysis for the results.

Author Response

#Reviewer 1

Comments and Suggestions for Authors

Comments:

The manuscript dealt with mathematical estimation of the energy, nutritional and health promoting values of multi-layer freeze-dried vegetable snacks. There are some questions needed to be addressed.

Abstract: Q: Please rewrite Abstract section. Abstract section is too descriptive. Please include appropriate information such as the methods used in the study and the important data and results obtained from the study.

Answer:

Abstract has been rewritten (Lines 10-12, 14-17 and 24-26).

Comments:

Lines 228-242 and lines 276-282: Q: Why the energy calculated using different units?

Answer:

Correction has been made to the manuscript text and units of all estimated parameters are now consistent in the paper (Lines 294-295, 302).

Comment:

Equation 4: Q: Please rewrite the equation using appropriate mathematical symbol.

Answer:

Corrections have been made to the manuscript text and symbols used in the mathematical equations are now consistent in the paper. 

Comment:

Tables 4-6: Q: Please provide the statistical analysis for the results.

Answer: The statistical analysis was added to Tables 4-6.

Reviewer 2 Report

The aim of the reviewed work is to assess the energy, nutritional and health-promoting value of freeze-dried plant products with hydrocolloids. The topics discussed are always topical as they concern the nutritional sphere and food production residues.

My comments regarding the content of the article:

  • The introduction is very extensive and the genesis of the purpose of the work is based on several lines. I propose to highlight and present the research purpose more.
  • The methodology does not specify how the nutritional and energy values of individual products were determined (laboratory equipment).
  • 3) The section on conclusions should present precise sentences resulting from the research. At the moment, it can be treated as some kind of summary.

Author Response

#Reviewer 2

Comments and Suggestions for Authors

Comments:

The aim of the reviewed work is to assess the energy, nutritional and health-promoting value of freeze-dried plant products with hydrocolloids. The topics discussed are always topical as they concern the nutritional sphere and food production residues.

My comments regarding the content of the article:

  • The introduction is very extensive and the genesis of the purpose of the work is based on several lines. I propose to highlight and present the research purpose more.

Answer:

This section has been improved to be more clear and consist (Lines 36-37, 39-40, 59-61, 71, 93-97, 103-104, 149-154, 157-165, 181-189, 239-240, 245-246).

Comments:

  • The methodology does not specify how the nutritional and energy values of individual products were determined (laboratory equipment).

Answers:

The methodology described mathematical methods of the nutritional and energy values estimation, which were the only ones used. Research presented in the manuscript did not consist of laboratory and analytical methods, but calculation based on tabular data and equations presented in the manuscript. To clarify this aspect we added some introduction to the 2.2 Computational methods section (Lines 284-290), as follows: “Estimation of the energy, nutritional and health-promoting value of the freeze-dried multi-layer vegetable snacks was determined on the base of tabular data about nutritional value of components contained in the formulation of the products [39-41]. For the purpose of the study, it had been established that one serving of each snack was one freeze-dried bar (10 g) and each layer shared 1/3 of the total product weight. All calculations were made in duplicate Excel 2019 (Microsoft, Redmond, Washington, United States of America).

Comments:

  • 3) The section on conclusions should present precise sentences resulting from the research. At the moment, it can be treated as some kind of summary.

 Answer:

Conclusions have been rewritten to be more clear and consist (Lines 815-851).

Reviewer 3 Report

General comments

The manuscript is well written in general and presents a very interesting and useful study. However, some points need to be improved before publication. The article is unnecessarily long and the organization must be reviewed. For example, in the methodology section, there is a lot of theoretical information. Finally, some important references are missing.

Abstract

  1. Please include a short introduction before mentioning the purpose of the study.

Introduction

  1. Line 138. “Hydrocolloids also have the ability to form gels. Thanks to these properties, they are used to create functional food”.

How can hydrocolloids create functional food? Please rephrase the sentence to avoid misunderstandings.

  1. Line 163. “It is estimated that 1/3 of food is wasted and lost as a result of unintentional or acci-163 dental activities in the world..”

Please add proper references.

  1. Line 171. Authors mention that the information is taken from FAO. However, it is necessary to include the reference so that the reader can expand the information from the source.
  2. What is the purpose of including in the introduction information regarding food waste and overproduction? The snacks will be made with agro-industrial waste?

Materials and Methods

  1. Line 220. Please change the word recipes to formulations all over the document.
  2. It is important mentioning the importance of the selected research material, why did you select this material instead another agricultural product?
  3. Line 224. Although another article is mentioned, it is necessary to include in this section information regarding plant material used, i.e geographical localization, weather conditions, and other relevant information.
  4. After Research material section, authors must include a section where explain the conditions under snacks were made. What was the treatment for the vegetables? disinfection? scalding? cut size? Also explain the main reason that the products have been made with those formulations.
  5. It section contains a lot of theoretical information, please provide information regarding only analysis conditions and keep only the most important information for the results section.

 Results and discussion

  1. Since some components in formulation are amylaceous products like potato, can you ensure that the manufacturing process of the snack guarantees the digestibility of the starch present in the potato?
  2. When mentioning Covered Daily Requirement CDR, for protein, fat, carbohydrates and energy in the human diet, specify the values and the source from information was taken.

Conclusions

  1. It may be improved in order to make it more impressive, and not only repeat the general information of the study. Please highlight the scientific conclusion.

Author Response

#Reviewer 3

Comments and Suggestions for Authors

Comments:

General comments

The manuscript is well written in general and presents a very interesting and useful study. However, some points need to be improved before publication. The article is unnecessarily long and the organization must be reviewed. For example, in the methodology section, there is a lot of theoretical information. Finally, some important references are missing.

Answers:

Introduction section has been shortened (Lines 36-37, 39-40, 59-61, 71, 93-97, 103-104, 149-154, 157-165, 181-189, 245-246) and the methodology section was improved (Lines 255-280, 284-290, 310-313, 322-335, 387-390). It is a pity that the reviewer did not indicate the missing citations, which we could add to the manuscript. Taking into account that the paper is very long we decided to not add new citations.

Comment:

Abstract

Please include a short introduction before mentioning the purpose of the study.

Answer:

The short introduction was added to improve this section (Lines 10-12).

Comments:

Introduction

Line 138. “Hydrocolloids also have the ability to form gels. Thanks to these properties, they are used to create functional food”.

How can hydrocolloids create functional food? Please rephrase the sentence to avoid misunderstandings.

Answers:

These sentence was explained by adding more explanation about the gelling property and the effects on the production of functional food (Lines 157-165), as follows: “Hydrocolloids also have the ability to form gels, which maintain their shape and structure after water removal due to freeze-drying, and that allows to manufacture at-tractive for consumer products. Moreover, their ability to water binding and swelling capacity make them food components responsible for keeping the feeling of satiety for a long time. As was mentioned before, hydrocolloids are high molecular weight biopolymers that act in human’s digestive system as dietary fiber. Thanks to referred properties of hydrocolloids, they are established as additives that can be used to create functional food.”

Comments:

Line 163. “It is estimated that 1/3 of food is wasted and lost as a result of unintentional or acci-163 dental activities in the world..”

Please add proper references.

Answer:

The proper citation was added to this part (Lines 193, 954-955), as follows: “35.     Vilariño, M. V., Franco, C., & Quarrington, C. Food loss and waste reduction as an integral part of a circular economy. Front. Environ. Sci., 2017, 5, 21.

Comments:

Line 171. Authors mention that the information is taken from FAO. However, it is necessary to include the reference so that the reader can expand the information from the source.

Answer:

The proper citation was added to this part (Lines 200 and 956), as follows: “36. Food and Agriculture Organization. Global Food Losses and Food Waste–Extent, Causes and Prevention, 2011, Rome: FAO

Comments:

What is the purpose of including in the introduction information regarding food waste and overproduction? The snacks will be made with agro-industrial waste?

Answers:

Yes, the snacks were developed as a part of the project that aimed to manage post-calibration outgrades generated during frozen vegetables manufacturing. Relevant information was included in the Material and methods section (Lines 255-270).

Comments:

Materials and Methods

Line 220. Please change the word recipes to formulations all over the document.

Answer:

The corrections have been done (Lines 249, 281, 409, 410, 498, 499, 584, 585, 586).

Comments:

It is important mentioning the importance of the selected research material, why did you select this material instead another agricultural product?

Answers:

The process of developing the snacks has been reported previously in the paper cited in this section (Line 254). The material used in this research was prepared with the use of raw materials supplied by co-participants of the project, and one of the main goals of the project was to manage post-calibration outgrades form already existing manufacturing process.

Comments:

Line 224. Although another article is mentioned, it is necessary to include in this section information regarding plant material used, i.e geographical localization, weather conditions, and other relevant information.

Answers:

The information about the manufacturer were included in Research material section (Lines 255-257). It is very valuable comment and we will consider it in further research, but in the case of this study more details about origin and the process of vegetables cultivation mentioned in the comment are not available for us to find.

Comments:

After Research material section, authors must include a section where explain the conditions under snacks were made. What was the treatment for the vegetables? disinfection? scalding? cut size? Also explain the main reason that the products have been made with those formulations.

Answers:

The technological methods used for preparation of the freeze-dried snacks were briefly described (Lines 257-280, 284-290). The process of developing the snacks has been reported previously in the paper cited in this section (Line 254).

Comments:

It section contains a lot of theoretical information, please provide information regarding only analysis conditions and keep only the most important information for the results section.

Answers:

Material and methods section has been corrected (Lines 255-280, 284-290, 310-313, 322-335, 387-390). Some of the irrelevant information was deleted and some transferred to the Results and discussion section, but, according to Applied Sciences requirements, methodology used in the research must be described in details, so we left those that seemed the most important.

Comments:

Results and discussion

Since some components in formulation are amylaceous products like potato, can you ensure that the manufacturing process of the snack guarantees the digestibility of the starch present in the potato?

Answers:

It is very valuable comment and we will consider it in further research, but in the case of this study more details cannot be added, since we did not estimate a degree of digestibility, however it will be a subject for further investigations.

Comments:

When mentioning Covered Daily Requirement CDR, for protein, fat, carbohydrates and energy in the human diet, specify the values and the source from information was taken.

Answer:

It was estimated based on the recommended in Poland amount of energy (2000 kcal) per person per day.

Comments:

Conclusions

It may be improved in order to make it more impressive, and not only repeat the general information of the study. Please highlight the scientific conclusion.

Answers:

This section has been rewritten (Lines 815-851).

Reviewer 4 Report

The manuscript "Mathematical estimation of the energy, nutritional and health-promoting values of multi-layer freeze-dried vegetable snacks" presents a very interesting argument on the novel food/snacks.

The authors have explained adequately the needs for preparing such types of snacks, and the introduction effectively has supported the idea based on the accomplished worked done by other researchers.

The applied methods have been completely well described and well designed which demonstrate that the authors have paid attention carefully not only to the physico/chemical aspects but also to the nutritional values.

The results are distinctly presented which supports the original idea and the accuracy of designated methods.

Overall, I found this work very interesting and useful, which presents a typical product with health-promoting ingredients, and it fits the sustainable development policy well.

I would like to suggest to the authors to complete their work with additional assessments on the taste tests of their products and consumer acceptance.

Author Response

#Reviewer 4

Comments and Suggestions for Authors

Comments:

The manuscript "Mathematical estimation of the energy, nutritional and health-promoting values of multi-layer freeze-dried vegetable snacks" presents a very interesting argument on the novel food/snacks.

The authors have explained adequately the needs for preparing such types of snacks, and the introduction effectively has supported the idea based on the accomplished worked done by other researchers.

The applied methods have been completely well described and well designed which demonstrate that the authors have paid attention carefully not only to the physico/chemical aspects but also to the nutritional values.

The results are distinctly presented which supports the original idea and the accuracy of designated methods.

Overall, I found this work very interesting and useful, which presents a typical product with health-promoting ingredients, and it fits the sustainable development policy well.

I would like to suggest to the authors to complete their work with additional assessments on the taste tests of their products and consumer acceptance.

Answers:

Thank you for your kind review. Sensory analysis has been conducted and the results are planned for publication as individual paper.

Round 2

Reviewer 1 Report

The authors have addressed all my concerns.